# Midwife-Led Mobile Antenatal Clinic: An Innovative Approach to Improve Utilization of Services in Pwani, Tanzania

**DOI:** 10.3390/ijerph21111446

**Published:** 2024-10-30

**Authors:** Beatrice E. Mwilike, Joanne Welsh, Kasusu K. Nyamuryekung’e, Alex J. Nyaruchary, Andrea B. Pembe, Mechthild M. Gross

**Affiliations:** 1Department of Community Health Nursing, Muhimbili University of Health and Allied Sciences, Dar es Salaam 11103, Tanzania; alex.nyaruchary@muhas.ac.tz; 2Maternal and Newborn Health, Cambridge CB24, UK; jovwelsh@gmail.com; 3Department of Community Dentistry, Muhimbili University of Health and Allied Sciences, Dar es Salaam 11103, Tanzania; kasusu.nyamuryekunge@muhas.ac.tz; 4Department of Obstetrics and Gynecology, Muhimbili University of Health and Allied Sciences, Dar es Salaam 11103, Tanzania; andrea.pembe@muhas.ac.tz; 5Midwifery Research and Education Unit, Hannover Medical School, 30625 Hannover, Germany; gross.mechthild@mh-hannover.de

**Keywords:** midwifery-led, antenatal care, mobile clinic, utilization, Sub-Saharan Africa

## Abstract

Participating in antenatal clinics is a major determinant in reducing poor maternal and neonatal birth outcomes. We aimed to evaluate the utilization of antenatal clinic (ANC) services provided by a mobile clinic led by skilled midwives and determine the acceptability in the Pwani region, Tanzania. For a year, the mobile clinic, nicknamed “Mkunga Kitaani” and equipped with necessary tools and staff, served seven villages in the Kisarawe district that lacked health facilities. The research was conducted using a descriptive study design, incorporating both qualitative and quantitative methods. Qualitative and quantitative data were collected through 12 interviews and 214 medical records among pregnant women, respectively. The results show that approximately 17% of the women initiated ANC early, while 36% made their visit during their third trimester. Participants generally preferred the mobile clinic over traditional facilities due to its provision of comprehensive care. However, challenges such as clinic unreliability during the rainy season and limited availability of tests, including obstetric ultrasounds, were noted. Despite hurdles, the study highlighted increased ANC access and community engagement, suggesting potential for expansion to other underserved rural areas. The findings underscore the importance of innovative approaches to ANC delivery in regions with limited healthcare infrastructure.

## 1. Introduction

Antenatal care (ANC) services serve as an essential healthcare strategy in improving maternal and neonatal health outcomes both in high and low- and middle-income countries. However, more efforts are needed to achieve the sustainable development goals (SDGs), that is by 2030, the global maternal mortality ratio (MMR) should be reduced to 70 per 100,000 live births [1]. Despite the global decrease in MMR by 34% between 2000 and 2020, the maternal death rate is still high and unacceptable. In 2020, 278,000 maternal deaths occurred with 87% of deaths occurring in Sub-Saharan Africa (SSA) and Central and Southern Asia, of which 70% (202,000) of maternal deaths occurred in Sub-Saharan Africa [2]. The observed maternal death rates are associated with preventable and treatable direct and indirect causes, including severe bleeding (mostly bleeding after childbirth) (27.1%), infections after childbirth (10.7%), high blood pressure during pregnancy (14.0%), pre-eclampsia and eclampsia (14.5%), complications from labour and birth (2.8%), or unsafe abortions (7.9%) [3,4].

There is well documented evidence that midwifery-led continuity of care models provide flexibility for women [5], improve satisfaction with care, and lead to fewer interventions than when a different model of care is offered [5,6]. Antenatal care is primarily the responsibility of midwives, although this varies between countries, and evidence suggests that continuity of care that is midwife-led has substantial benefits for the woman and her baby, including better outcomes and improved satisfaction [6]. ANC services that are accessible to all is one of the priorities in reducing adverse maternal and neonatal health outcomes, up to and including death.

According to the World Health Organization [7], the well-being of pregnant women and their newborns depends on adhering to at least four recommended ANC visits. However, inequalities in access to maternal health care, including ANC, remain a profound challenge among women living in rural areas compared to urban areas [1]. In high-income countries like the United States and Canada, ANC utilization is generally much higher, with over 80–90% of pregnant women accessing care during their pregnancies [7]. However, in Sub-Saharan Africa, the utilization of ANC services between 2010 and 2018 was poor; approximately 13% of women did not utilize ANC during the pregnancy period while 35% only partially utilized the service [8]. According to the Tanzania Demographic Health survey among women aged 15–49 who had a live birth or stillbirth three years before the survey, 34% of women started ANC within the first three months of pregnancy, and 6% did not seek ANC until the seventh month or later [9].

Moreover, geographical location contributes to the utilization of ANC services. Rural areas often face significant challenges in terms of accessibility and availability of healthcare services with the distance to healthcare facilities, lack of reliable transportation, and limited infrastructure posing significant barriers for rural women, and making it difficult for them to attend ANC regularly [10,11,12,13]. Additionally, the socioeconomic status of rural women, which may be lower compared to their urban counterparts, can further exacerbate the challenges they face in utilizing ANC healthcare services [14].

To address accessibility challenges and improve maternal healthcare outcomes in rural communities, innovative solutions are needed. Mobile ANC serves as a backup to the utilization of ANC services in rural communities where women benefit from ANC services in their communities. These clinics, which reach out to communities often neglected by traditional healthcare systems, offer a vital service. They ensure that expectant mothers receive essential antenatal care, contributing to the health and well-being of both the mothers and their unborn children. This well-known approach to healthcare delivery has proven to be a lifeline for those residing in areas where medical facilities are scarce or inaccessible [15].

Mobile clinics not only bring healthcare services to the doorsteps of those in need but also embody a compassionate extension of healthcare equity in conflict affected areas [16,17]. Current evidence indicates women using mobile ANC services feel more comfortable and empowered to engage with healthcare providers, leading to increased trust and adherence to ANC recommendations [16]. The absence of mobile clinics in the Kenyan setting was shown to have a negative effect on the utilization of maternal and child health services [12]. However, when comparing the quality of care between mobile and fixed ANC services, a study conducted in Haiti found no difference in the care provided in the two ANC models, observing low quality of care in both settings [18].

Mothers living in remote and underserved regions have expressed high levels of satisfaction with the antenatal care provided by mobile clinics [15]. In India, among pregnant women in rural and hard-to-reach settings, mobile clinics were found to be a feasible and acceptable model to deliver healthcare, including ANC services, with a total of 76% of 2211 women in the 144 villages receiving antenatal care [19]. Furthermore, in Tanzania, between 2008 and 2012, Ngorongoro’s outreach clinic network increased the accessibility of antenatal care in a rural area [20]. Regarding the cost associated with the utilization of services at mobile health clinics, pregnant women reported having missed their scheduled visit due to lack of money (15%) and time (9%) [21]. Despite the challenges of running mobile clinics to cover the gap in providing ANC services, we implemented a midwife-led mobile antenatal clinic in the Pwani Region in Kisarawe district between 2021 and 2022.

This study was designed to evaluate the utilization of antenatal clinic (ANC) services provided by a mobile clinic led by skilled midwives and determine the acceptability among women residing in rural areas.

## 2. Materials and Methods

This was a community-based operational research project whereby pregnant women were recruited to receive antenatal care services through a mobile antenatal clinic. The mobile clinic services were provided within the selected hard-to-reach villages in Kisarawe district within the Pwani region of Tanzania. The research was conducted using a descriptive study design, incorporating both qualitative and quantitative methods for data collection and analysis. The qualitative approach was used by conducting in-depth interviews with participants, allowing for detailed insights into their perspectives on using the mobile clinic services. The quantitative approach involved collecting and analysing numerical data from medical records (214 records) drawn from those women experiencing mobile ANC service utilization.

### 2.1. Study Setting

The study was conducted in Pwani region, Tanzania, which is divided into six administrative districts. The Kisarawe district is one of these districts and was the selected intervention site based on remoteness and population to fit the study purposes. Despite several efforts made by the government of Tanzania and other partners to reduce maternal and newborn deaths, maternal and neonatal mortality has remained a serious problem in the Pwani region, including the Kisarawe District. According to DHIS 2 data from 2015 to 2019, the MMR for Kisarawe district ranges from 28:100,000 to 356:100,000 live births.

Kibuta ward is one of the areas facing barriers in the provision of quality maternal health care. The ward has a total of nine villages and only two villages have a health facility at the dispensary level. The remaining seven villages do not have any health facilities, and this means women have to travel long distances to access health services. The major challenges that hinder access to antenatal care in the intervention sites include inadequate health facilities, lack of equipment, and poor infrastructure. Furthermore, long distances from residential areas to facilities and the lack of reliable transportation options leave women in the area struggling to reach the services.

### 2.2. Participants’ Inclusion and Exclusion Criteria

The study included all pregnant women residing in the Kibuta ward during the study period from September 2021 to June 2022. Both of these wards are located in remote rural areas. The total population of Kibuta ward was 10,201 people and there were 2293 women of reproductive age. Pregnant women with a high-risk pregnancy were excluded. However, they were referred to the nearby facility for further management.

### 2.3. The Mobile Antenatal Clinic

One 4-wheel-drive mobile clinic was designed to provide, as much as possible, the work capabilities of a basic standard antenatal clinic in the selected ward. The vehicle was used as transport and provided some working space for midwives. It contained a well-equipped clinic and a mini-laboratory capable of providing commonly needed tests. The vehicle for the mobile clinic carried a medium-sized tent, an examination bed, drawers for the storage of instruments and consumables, and a cool box for the storage of vaccines. The vehicle also included a waste disposal tank designed for hazardous materials. The mobile clinic was operated by two trained and skilled midwives at diploma and degree levels of education. The vehicle would park in an area convenient for women to access within a village following the planned schedule. The community health workers (CHWs) worked closely with the mobile clinic midwives to ensure women were informed of the mobile clinic visit and coordinate the meeting places.

The mobile clinic service operation started in September 2021 and ran until June 2022. During implementation, each village was visited one day per week on a regular schedule. The mobile clinic was scheduled to depart from the district hospital area at 06:00 each day to provide health care services for the assigned village from 08:00–09:00 h until 15:00 h. The vehicle would park either at a school compound, village office compound, or an open space within a particular village depending on the convenient accessibility spot within the village. Conditions during the rainy season meant there were several occasions where the mobile antenatal clinic was unable to be held on planned dates, leading to a certain level of unreliability.

Clinical care was carried out following the WHO (2016) recommendations for antenatal care and the National Antenatal Care Guideline (2019). The model suggests eight contacts with a healthcare provider throughout the pregnancy, with specific interventions and assessments at each contact. The first contact should be in the first trimester, followed by two contacts in the second trimester and five contacts in the third trimester. The model also covers nutrition, prevention and treatment of common physiological symptoms, and counselling and support for women who may face intimate partner violence or other challenges.

During antenatal visits at the mobile clinic, women would receive nearly all the required services according to the policies and guidelines free of charge, with the exception of ultrasound scans. For each contact, we were able to carry out investigations while ensuring all safety and infection prevention measures were taken into consideration. One midwife was responsible for registration, taking blood pressure, weight, and height measurements and providing health education. Another midwife was responsible for the physical assessment of the pregnant woman, fetal assessment, and blood investigations.

Obstetrical danger signs were also assessed during the clinic. Obstetric danger signs are symptoms or conditions that indicate a serious risk to the health or life of a pregnant woman or her baby. They require immediate attention and referral to a higher level of care. Some of the most common obstetric danger signs are vaginal bleeding, severe headache, blurred vision, convulsions, fever, severe abdominal pain, and difficulty in breathing [22,23].

The laboratory procedures carried out at the mobile antenatal clinic included all standard tests conducted during pregnancy, including checking of hemoglobin, blood sugar, urine check using dipstick, and counselling and testing for HIV and syphilis. The clinic also provided iron and folic acid supplements, antihelminthics, prophylactic antimalaria (Sulphadoxine Pyrimethamine), and insecticide-treated nets as prophylactic measures. Furthermore, women received counselling on individualized birth planning and family planning based on their needs.

### 2.4. Data Collection

Data collection commenced from September 2021 and lasted until June 2022. Women who had a positive pregnancy were eligible to take part in the study. Women who accessed antenatal care in the mobile clinic were recruited to take part in the study after providing consent. We used an antenatal card number four (Reproductive and Child Health card number 4-RCH4) to record the progress of pregnancy. The first contact was recorded, and women were provided with an appointment date for their subsequent ANC visit, taking into account the ANC contact schedule. The information was also recorded in respective national registers (Mfumo wa Taarifa za Uendeshaji wa Huduma za Afya-MTUHA book number 12) as per protocols and women were encouraged to attend a health facility for labor and birth.

The information recorded on the antenatal care register was then entered into a file developed using Research Electronic Data Capture (REDCap) software, which is an electronic record developed to store data daily. The midwives in the mobile clinic were responsible for collecting these data and uploading them into the REDCap software version 11.1.5. The project data manager was responsible for performing data quality checks on a weekly basis. The details recorded in the ANC register and the electronic file included, but were not limited to, date of visit/contact, number of ANC visits, gestational age, gravidity, parity, number of babies alive, number of abortions, hemoglobin level, mother’s age, report on vaccination, date of follow-up visits, and sexually transmitted infections (STIs) and Prevention of Mother To Child Transmission (PMTCT) services records.

In-depth interviews were conducted using a structured interview guide to gather qualitative data. These interviews were conducted with women who had received services from the mobile clinic. Twelve women, selected with purposeful sampling, participated in these interviews. All interviews were conducted by an experienced research assistant in Kiswahili, audio-recorded, lasted 10–20 min, and stopped at the 12th participants after realizing that no new information was extracted from repeated interviews. The interview guide comprised questions that explored the women’s perspectives on utilizing the mobile antenatal clinic. It also sought to compare their experiences with previous antenatal care received at traditional health facilities and to identify any disadvantages or challenges of using the mobile clinic.

### 2.5. Data Analysis

Quantitative data were analyzed using SPSS statistical software package version 25. Standard descriptive statistics were applied in the analysis. Continuous variables were described in mean and standard deviation (SD). The categorical variables were characterized by absolute and relative frequencies.

Qualitative data analysis was conducted through thematic analysis. The analysis process started with the familiarization of the data, where the researchers immersed themselves in the details and nuances of the content. This was followed by generating initial codes. Furthermore, the codes were sorted into potential themes, which were refined into a coherent pattern that accurately reflected the dataset. The next step involved reviewing themes, ensuring they were representative of the data and formulating a clear definition and name for each theme. The final step was the production of the report, which communicates the findings of the analysis.

### 2.6. Ethical Considerations

The ethical approval for the study was sought from the Research and Ethics Committee of the Muhimbili University of Health and Allied Sciences (MUHAS) with approval number Ref. No.DA.282/298/01.C/. The relevant authority granted permission to conduct the study, and the Kisarawe District Commissioner launched the project. The midwives providing care in the mobile clinic sites explained the study to pregnant women and responded to any questions women had about the study. If a woman decided to participate in the study, she was asked to complete a consent form. If a woman wished to use the mobile antenatal clinic for care during her pregnancy but did not wish to take part in the study, she was still able to receive her antenatal care at the mobile antenatal clinic. Women below 18 years old were also informed and consent was obtained from their parents/guardians. Participants were informed that their identity would be pseudonymized but that any data generated would be linked to the Kisarawe district health management and information system as per protocols. The data generated were treated in a confidential manner exclusively for scientific purposes. Only the principal investigator, co-investigators, and project advisors were authorized to handle and examine the data.

## 3. Results

### 3.1. Socio-Demographic Information

A total of 214 participants utilized a mobile antenatal clinic and took part in the study. The majority of the women were young, with 85.5% of them being between 15 and 34 years old. The mean age was 26.3 years, with a standard deviation of 6.4 years. Most of the women were married (70.6%), followed by cohabiting (15.4%) and unmarried (13.6%). The level of education was low among the women, with 71.5% of them having primary education or less. Only 13.6% of them had completed secondary education or higher. More than 50% were housewives, and peasants accounted for 34.1% of the women, while self-employed and employed were 14% and 1%, respectively. Table 1 provides additional details.

Women reported that prior to the introduction of the mobile antenatal clinic, they had to travel distances ranging from 2 km to 17 km to reach the healthcare facility. Most women reported needing to travel a long distance and that they would spend one to two hours walking distance to reach the health facilities. The only reliable transport was a motorcycle (bodaboda) and they would pay a total of TZS 6000 to 24,000 for a round trip. The distance to the facility and high cost of transport made it difficult for pregnant women to seek antenatal services accordingly.

### 3.2. Timing and Number of ANC Contacts

More than half of the women were multigravida (58.9%), meaning they had been pregnant two to four times. Primigravida (21%) and grand multigravida (20.1%), meaning being pregnant for the first time and being pregnant for more than four times, respectively, were almost equally distributed. The timing of ANC booking was late for most of the women, with only 16.8% of them booking in the first trimester. Nearly half of them booked in the second trimester (47.7%) and more than a third booked in the third trimester (35.5%). Furthermore, the number of ANC contacts was recorded, whereby it was revealed that only 20.4% of women were able to attend four or more ANC contacts and 0.46% had eight contacts with the midwives. The cumulative total number of ANC contacts made at the mobile clinic during the study period was 579. Table 1 provides details regarding the timing of ANC bookings in gestational age (GA) weeks and Figure 1 provides the number of visits for each ANC contact.

### 3.3. Danger Signs and High-Risk Detection

During the mobile clinic service provision, we were able to assess obstetric danger signs and detect high-risk pregnancies. The majority of women in the study experienced severe headache (*n* = 66) and fever (*n* = 61) as danger signs during pregnancy. Other symptoms, such as difficulty in breathing (*n* = 15) and blurred vision (*n* = 13), were also observed, as shown in Figure 2. These signs suggest the potential presence of malaria infection, anemia, or hypertensive disorders, which pose serious risks and may lead to complications for the mother and the newborn.

We conducted blood tests for malaria, HIV, syphilis, and hemoglobin levels to screen for infections or anemia among the women. We measured the hemoglobin level at each visit and out of 579 tests performed, we recorded 251 cases of mild anemia (10–10.9 g/dL), 92 cases of moderate anemia (8.6–9.9 g/dL), and 10 cases of severe anemia (<8.5 g/dL). We also detected mild-moderate pre-eclampsia in about 2% and severe pre-eclampsia in about 1% of the women. Out of the total number of visits, 82 were positive for malaria infection. In addition, we found that about 20% of grand-multipara and 12% of HIV-positive women had high-risk pregnancies. We provided treatment at the mobile clinic for mild to moderate malaria cases and referred the severe cases to the nearest health facility.

### 3.4. Women’s Perspectives for Using Mobile ANC Clinic

A total of 12 women shared their perspectives regarding using the mobile services through in-depth interviews. About 50% were aged 25-34 years, 50% married and 66.7% had completed primary education. The characteristics are displayed in Table 2. The themes and key qualitative findings are shown in Table 3.

#### 3.4.1. Service Comparison with the Primary Healthcare Facility

The participants who previously experienced the usual care from the primary healthcare facility were asked to compare the quality of services they received with mobile ANC clinic services. The participants described contrasting experiences at the two different care points. The majority responded that they found the mobile clinic services to be better than the services they previously received in the facilities. At the health facility, the participants felt that the care was inadequate and some of the required tests were missing; sometimes they did not have their hemoglobin level tested nor were they provided with medication to potentially increase their hemoglobin levels. However, their experience at the mobile clinic was markedly different. Here, they received comprehensive care, including hemoglobin level testing and access to other medical tests.

“*When I attended there (facility x)...mmm... I didn’t even measure the amount of blood (mmh) I wasn’t even given pills to increase blood...mmm...but when I arrived here (mobile clinic) I got everything … and all other tests are available here unlike there...*”(P2)


*“…the good thing we see is that they (mobile clinic) have tests that were not available at that dispensary…”*
(P12)

#### 3.4.2. Challenges While Using the Mobile Clinic

The participants showed concern regarding some challenges they encountered despite having the mobile clinic in their villages. One of the reported challenges is the mobile clinic’s inability to provide ultrasound services and conduct blood grouping and cross-matching tests for each pregnant woman during the antenatal care contact.


*“...the issue of ultrasound for us it is difficult to get it…”*
(P7)


*“The service is good, but there is just one thing…blood group testing is not included, so we were asking for it to be added so that we can test everything. For the blood group, we are told to go and get tested at the health center.”*
(P8) 

Furthermore, the condition of the roads deteriorates significantly during the rainy season. For women travelling to the ANC at a health facility, this poses substantial challenges, sometimes resulting in them missing appointments. Similarly, the rainy season also impeded the ability of the mobile clinic to attend the villages on planned clinic days, meaning the clinic schedule was frequently changed during rainy season. As a result, women were not able to consistently attend for ANC even when services were being brought to their village.


*“...our challenge is the paths, …you may find a day like today dated thirty first if it rains some of our paths are impassable and therefore we are supposed to reschedule the clinic and therefore we struggle on where to pass.”*
(P6)

## 4. Discussion

We evaluated the utilization of ANC services provided by a mobile clinic led by skilled midwives and determined the feasibility and acceptability of antenatal care services. ANC is a critical component of maternal and child health care. Timely ANC booking is essential for ensuring optimal health outcomes for both pregnant women and their babies. Inappropriate timing of the first ANC booking can lead to adverse events such as perinatal death, stillbirth, and early neonatal death. Furthermore, the quality of ANC provision matters for positive birth outcomes and maternal satisfaction.

Our findings revealed that only a small fraction, 16.8%, initiated ANC during the first trimester, which is crucial for early detection and management of pregnancy-related complications. According to the World Health Organization (WHO), every pregnant mother should start booking ANC within the first 12 weeks of gestational age [24]. It is possible that late initiation of ANC can be attributed to the fact that the women accessing the mobile ANC were already pregnant when the mobile service began. Indeed, we observed that women who fell pregnant during the period in which the mobile clinic was available tended to initiate ANC earlier. Women initiated antenatal contact earlier when the mobile clinic was more established. The observed late ANC bookings and low attendance of recommended ANC contacts are consistent with findings from various other studies conducted in different regions and settings. Studies from Ethiopia reported similar patterns of delayed ANC initiation, with only a small percentage of women initiating ANC in the first trimester, and the majority delaying until the second or third trimester [25,26,27]. These studies have suggested reasons for delayed ANC to include distance to reach the health facility and lack of knowledge regarding the importance of early ANC initiation [27]. Women living in rural areas tend to book ANC later than their urban counterparts [24]. This suggests that in order to improve ANC timing, we need to scale up mobile clinic services in hard to reach areas as well as increase awareness through educational materials and campaigns [27] on the importance of early initiation of antenatal care seeking.

Additionally, the data revealed that only 0.46% of our sample attended the recommended number of ANC appointments, which is typically eight contacts throughout pregnancy [28]. This means that a significant proportion of pregnant women in our study did not receive the recommended number of ANC contacts. This could be attributed to women delaying initiation of ANC and therefore attending much less than expected. This may be because several women were already pregnant when the mobile clinic became available, meaning that we could not offer care to them from the beginning of their pregnancy. Furthermore, due to difficulties in the mobile clinic being able to access the villages during the rainy season, the clinic schedule was not always reliable and may have contributed to women having a reduced number of attendances. These findings were similar to studies in Ethiopia [29], Tanzania [21], and Sub-Saharan Africa (SSA) [30]. Adequate ANC contacts allow healthcare providers to monitor maternal and fetal health, provide essential education and support, and detect and manage any complications that may arise. To ensure women are aware of the recommended schedule for antenatal care and the benefits of this, more sensitization is needed.

Our findings also highlight the critical role of early detection and management of obstetric danger signs. The high incidence of severe headaches and fever reported are indicative of underlying conditions such as preeclampsia or malaria infections, which are significant contributors to maternal morbidity and mortality. This is evident as compared to findings from a study conducted in Mozambique, whereby there was an association of subclinical malaria, as well as of pre-eclampsia/eclampsia, in pregnant women with adverse maternal and neonatal outcomes [31]. The presence of other symptoms like difficulty in breathing and blurred vision, although less frequent, also point towards serious health concerns that could compromise pregnancy outcomes. Therefore, a thorough assessment of obstetric danger signs is paramount for the prevention of severe complications and death.

Our findings further highlight the acceptability of services provided at the mobile clinic versus those provided at the usual health facility. They suggest that while mobile clinics are an innovative approach to increasing the coverage of ANC, the quality of care provided was perceived to be better when compared to usual health facilities, with mobile clinics providing comprehensive services such as laboratory examinations. Our findings slightly differ from a study conducted in central Haiti [18], where there were notable differences between fixed and mobile clinics, with fixed clinics performing better in certain aspects of care such as intake and laboratory examinations, while mobile clinics excelled in providing supplies, iron-folic acid supplements, and Tetanus Toxoid vaccines. Findings from our study highlighted that women were disappointed that ultrasound scanning was not available in the mobile antenatal clinic. National recommendations for ANC in Tanzania include that every woman should have at least one ultrasound scan in pregnancy. However, ultrasound scanning is only offered within health facilities from the health center level. While lack of ultrasound scanning has been a shortcoming of the mobile antenatal clinic, this omission in care reflects a national level gap in care that requires substantial investment in training and equipment to resolve.

This study adds to the body of knowledge on the challenges that women living in hard-to-reach areas encounter, even in the presence of the mobile clinic. The unpredictability of clinic schedules due to impassable roads can lead to a lack of trust and discontinuity in care, as patients are unsure of when services will be available. This uncertainty may discourage individuals from seeking care. Where mobile ANC services are offered, there needs to be a robust plan in place to combat inconsistences in clinic schedules due to effects of rainy season. Despite these challenges, evidence reveals that mobile clinics extend healthcare access to vulnerable populations at a fraction of the cost of running a traditional hospital or care facility [14,32]. In response to these challenges, there is a need for innovative solutions that ensure the reliability of mobile clinics regardless of weather conditions. This could include the use of more robust vehicles capable of navigating difficult terrains, or the development of alternative transportation methods during the rainy season.

Future studies should attempt to replicate our innovation on a larger scale and among diverse populations of pregnant women residing in hard-to-reach areas. The establishment of mobile antenatal clinic services can help improve access to vital healthcare services for pregnant women together with early detection and prevention of complications. We are further recommending a study to analyze the cost-effectiveness of mobile clinics in delivering antenatal care compared to stationary health facilities.

## 5. Conclusions

Approximately 17% of participants were able to initiate antenatal care (ANC) services early, and the majority reported receiving superior care through the mobile clinic compared to traditional health facilities. Despite this, challenges were noted, including unreliable mobile clinic schedules during the rainy season and the absence of certain tests, such as blood grouping and ultrasound examinations. Our research demonstrates enhanced access to and utilization of ANC services alongside increased community engagement. These findings suggest the potential for expanding the project to other rural areas that are difficult to reach and have limited ANC services available.

## Figures and Tables

**Figure 1 ijerph-21-01446-f001:**
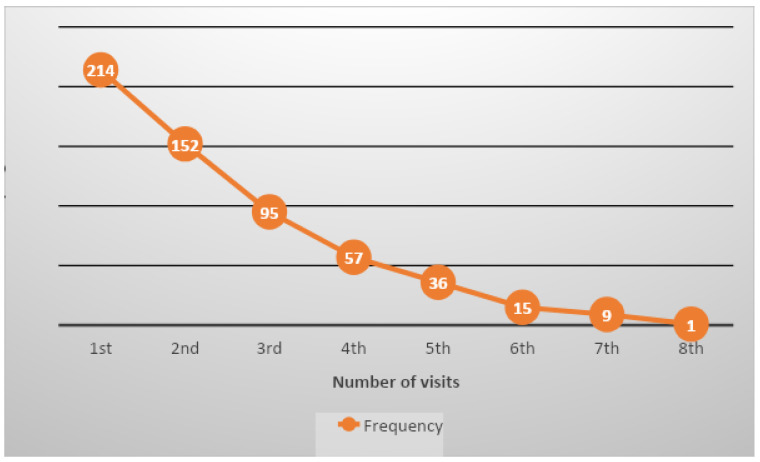
Number of ANC visits among pregnant women for each ANC contact.

**Figure 2 ijerph-21-01446-f002:**
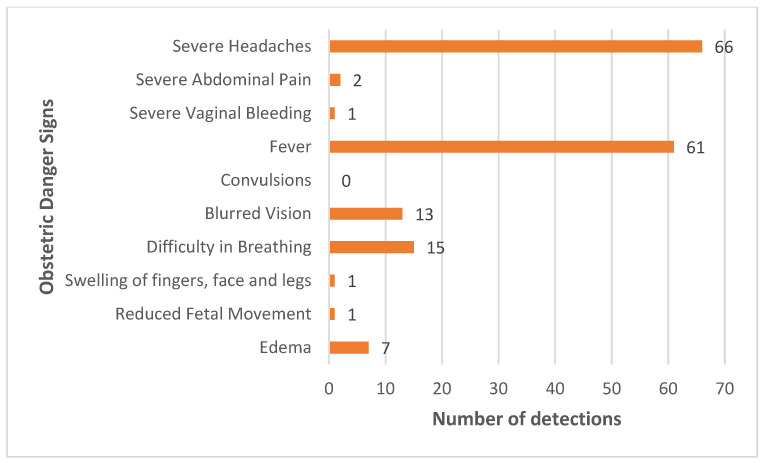
Number of women detected with obstetric danger signs.

**Table 1 ijerph-21-01446-t001:** Socio-demographic information and timing of ANC contacts (*n* = 214).

Variables	Frequency	Percentage
Age in years		
[Mean; SD]	[26.3; 6.4]	
15–24	100	46.7
25–34	83	38.8
35+	31	14.5
Marital status		
Married	151	70.6
Unmarried	29	13.6
Cohabiting	33	15.4
Separated	1	0.5
Education level		
No formal education	24	11.2
Primary incomplete	32	15.0
Primary complete	129	63.0
Secondary incomplete	6	7.5
Secondary complete	13	6.1
Current occupation		
Self-employment	30	14.0
Employed	2	1.0
Peasants	73	34.1
Other (housewife, student)	109	50.9
Gravidity		
Prime gravida	45	21.0
Multigravida (2–4)	126	58.9
Grand multigravida (5+)	43	20.1
Timing of ANC booking		
1st trimester (1–12 GA in wks)	36	16.8
2nd trimester (13–24 GA in wks)	102	47.7
3rd trimester (25–40 GA in wks)	76	35.5

**Table 2 ijerph-21-01446-t002:** Participant characteristics for qualitative interviews (*n* = 12).

Variables	Frequency	Percentage
Age in years		
15–24	4	33.3
25–34	6	50.0
35+	2	16.7
Marital status		
Married	6	50.0
Unmarried	2	16.7
Cohabiting	4	33.3
Education level		
No formal education	3	25.0
Primary complete	8	66.7
Secondary complete	1	8.3
Gravidity		
Prime gravida	2	16.7
Multigravida (2–4)	5	41.7
Grand multigravida (5+)	5	41.6

**Table 3 ijerph-21-01446-t003:** Summary of thematic areas and key findings.

Themes	Key Findings
A. Service Comparison Subthemes Quality of services received This theme examined the quality of ANC service received from the usual health facility compared to the service received from the mobile clinic among pregnant women who had received services from both points of care.	‑ Inadequate services from the usual health facility point of care. ‑ Comprehensive services at the mobile clinic. ‑ Having all the required tests at the mobile clinic that were not available at the facility, for example blood testing and urine checks.
B. Challenges at the Mobile Clinic Subthemes Lack of some tests Change of ANC schedule This theme explored the challenges the women encountered during mobile clinic usage.	‑ Lack of blood grouping check. ‑ Lack of ultrasound check. ‑ Women report changing mobile clinic schedules due to the inability of the vehicle to reach the village, especially during the rainy season when the roads are impassable.

## Data Availability

Datasets used and/or analyzed during the current study are available from the first author on request.

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
