# Peer review of "Midwife-Led Mobile Antenatal Clinic: An Innovative Approach to Improve Utilization of Services in Pwani, Tanzania"

_ijerph, 2024, doi:10.3390/ijerph21111446_

Round 1

Reviewer 1 Report

Comments and Suggestions for Authors

Abstract

Indicate the research method and design, and the sampling method used in your study.

Introduction

Good, however, it is silent about the comparison of care between mobile and usual health facility.

Materials and Methods

Indicate the research method you utilized in this study.
Indicate how qualitative and quantitative were used.
What was the rational of using both methods?
It is not clear whether you used mixed methods or qualitative and quantitative separately.  Please clarify
Indicate the research design used.

Participants’ inclusion and exclusion criteria

Line 127:What was the total of the target population?

Data collection

Line 195-196: When was the data collected?
How did you recruit the participants?
What about quantitative data?
Who conducted the interviews?

Line 197: Indicate why you selected 12 participants

Data analysis

Line 205: Under data collection there is no collection of quantitative data. It cannot be analyzed while it was not collected.
It is not clear whether quantitative data also revealed the themes.

Ethical considerations

Line 219: Why did the midwives explained the study to the pregnant women? It was indicated no were that assisted with data collection.

Line 221: Please recheck the meaning of the statement.

Line 226:How did you access the parents/guardians to sign the consent form?
Please add an 's' to make it plural since you used 'their' before these terms

Results

Line 253: spelling

line 278:  Is grand multi-gravida and HIV reactive pregnancies not high risk by nature? Please recheck

Line 289:  Is hemoglobin the blood volume?

Line 384: The comparison between mobile clinics and usual heath facility appears no were except here under the discussion.  Under, introduction and presentation of the results, there is no comparison. In addition, the study is not clear about which results (qualitative or quantitative) revealed what and what not revealed from the qualitative and quantitative. Did you merge the result? Please clarify

Line 484: please be consistent with the use of upper and lower cases.

Thank you

Comments on the Quality of English Language

Minor editing is required.

Reviewer 2 Report

Comments and Suggestions for Authors

This study aimed to evaluate the utilisation and satisfaction of antenatal clinic services by pregnant women accessing a midwife-run mobile clinic in Pwani, Tanzania. The authors used data from the antenatal care register and structured interviews with pregnant women to determine whether the mobile clinic increased utilisation and satisfaction of antenatal clinic services. A total of 214 pregnant women participated in the study, with only 17% meeting recommendations of accessing antenatal clinic services in the first trimester. Participants deemed the mobile clinic more acceptable than usual health facilities as care was more comprehensive. However, logistical issues (e.g., road closures) made the mobile clinic inaccessible at times. This is one of few studies aiming to improve utilisation of antenatal clinic services in Sub-Saharan Africa and provides evidence on the acceptability and feasible of implementing mobile antenatal clinics. Overall, I read this paper with great enthusiasm and deem it publishable by IJERPH, but I recommend it undergo minor revision before publication. I hope the authors will consider the following questions/comments and address them when necessary.

General Concept Comments: The aim of the study is vague and needs further clarification. As it is currently written, I interpreted the aim to be an evaluation of the mobile clinic at increasing the utilisation and satisfaction of antenatal clinic services. The results presented do not allow for conclusions to be made regarding this aim, which would require a controlled design with a comparison group. However, an aim that is to determine the utilisation (i.e., the quantitative data) and acceptability (i.e., the qualitative data) of the mobile antenatal clinic would be an appropriate alternative.

Specific Comments: While the manuscript overall was well-written, it would benefit from another round of proofreading. I have included some suggestions to improve the readability/formatting of the manuscript in the attached copy of the paper.
